# A Review into the Effectiveness of Ozone Technology for Improving the Safety and Preserving the Quality of Fresh-Cut Fruits and Vegetables

**DOI:** 10.3390/foods10040748

**Published:** 2021-04-01

**Authors:** Rinaldo Botondi, Marco Barone, Claudia Grasso

**Affiliations:** Department for Innovation in Biological, Agro-Food and Forest Systems, University of Tuscia, 01100 Viterbo, Italy; marco.barone95@hotmail.it (M.B.); claudiagrasso93@libero.it (C.G.)

**Keywords:** minimal processed, ready to eat, O_3_, microbial contamination, food properties

## Abstract

In recent years, consumers have become increasingly aware of the nutritional benefits brought by the regular consumption of fresh fruits and vegetables, which reduces the risk of health problems and disease. High-quality raw materials are essential since minimally processed produce is highly perishable and susceptible to quality deterioration. The cutting, peeling, cleaning and packaging processes as well as the biochemical, sensorial and microbial changes that occur on plant tissue surfaces may accelerate produce deterioration. In this regard, biological contamination can be primary, which occurs when the infectious organisms directly contaminate raw materials, and/or by cross-contamination, which occurs during food preparation processes such as washing. Among the many technologies available to extend the shelf life of fresh-cut products, ozone technology has proven to be a highly effective sterilization technique. In this paper, we examine the main studies that have focused on the effects of gaseous ozone and ozonated water treatments on microbial growth and quality retention of fresh-cut fruit and vegetables. The purpose of this scientific literature review is to broaden our knowledge of eco-friendly technologies, such as ozone technology, which extends the shelf life and maintains the quality of fresh produce without emitting hazardous chemicals that negatively affect plant material and the environment.

## 1. Introduction

### 1.1. Fresh-Cut Produce Overview

Fruits and vegetables are essential components of a healthy, balanced diet, as they are important sources of both macro and micronutrients. In fact, regular fruit and vegetable consumption is known to reduce the risk of several degenerative cardiovascular diseases [1,2] as they are rich in natural antioxidants with therapeutic properties. Consumers expect safe, high-quality foods that meet their individual dietary needs [3] and acknowledge the need to include “nutrient dense” fruits and vegetables rich in fiber, vitamins and minerals as well as antioxidants and phytochemical compounds in their daily diet [4,5,6,7,8,9].

According to European Community standards, fresh-cut fruits and vegetables, which are essential to human health and wellbeing, are defined as minimally processed, ready to eat and fresh-cut products, i.e., subject to reduced technology for direct consumption without further handling or with minimal handling [10].

Some examples of ready-to-eat (RTE) fresh-cut produce are broccoli and cauliflower florets, riced cauliflower, cut celery stalks, cut melon, peeled baby carrots, salad mixes, sectioned grapefruit, shredded cabbage, shredded lettuce, sliced bell peppers, sliced mushrooms, sliced pineapple, sliced tomatoes and cut zucchini squash, while cubed butternut squash and cubed sweet potatoes are not ready-to-eat (NRTE) fresh-cut produce [10].

Minimally processed fresh-cut fruits and vegetables are foods that have been washed, peeled and cut into 100% usable products, which are then packaged into retail-size PET, PVC and PP polymers containers with selective gas permeability, sometimes associated with modified atmosphere (Figure 1) [11].

According to the Produce Marketing Association (PMA, 2014), the US market for fresh-cut vegetables and fruit is estimated at USD 27 billion per year with bagged salads accounting for 61% of the market, other fresh-cut vegetables 27% and fresh fruit 11%.

The global marketing of such minimally processed fruits and vegetables in 2017 was USD 245.97 billion and is projected to reach USD 346.05 billion by 2022 (PMA, 2019). Fresh-cut produce accounts for 16% of total fruit and vegetables sales. It is difficult to estimate the post-marketing losses of fresh-cut produce; however, given the high perishability of fresh-cut produce compared to intact produce, the total retail loss caused by wastage of fresh-cut produce may exceed USD 9–10 billion.

For at least two decades, the European fresh-cut market has shown a clear preference for fresh-cut, packaged lettuce, which represents approximately 50% of the total fresh produce market volume, while only 10% is represented by fresh-cut fruit and the remaining 40% by ready-to-cook and crudités products. More specifically, between 2009 and 2014, a significant increase of approximately 19% was observed in fresh-cut fruit consumption, although it is only a small percentage of the overall fruit and vegetable market (approximately 10%). The countries with the highest fresh-cut produce growth rates are the Western European nations (Spain, Germany, France, Italy, the Netherlands), while fresh produce consumption growth in southeastern European countries is still negligible [12]. In addition to the abovementioned markets, fresh fruit snack packs have gradually gained popularity in several countries, yet only to a limited extent in China (freshplaza.it).

### 1.2. Safety and Sanitisation Practices

Several food commodities, including fresh-cut produce, have been associated with foodborne illness outbreaks, and many fresh-cut products are consumed raw. Other foods are cooked and safe to eat because heat destroys pathogenic microorganisms.

Preventive controls to significantly minimize the risk or prevent the contamination of fresh-cut produce with pathogens are essential food safety practices [13]. These controls are generally carried out in compliance with good manufacturing practice (GMP), to identify the critical control points of the HACCP system in accordance with the industrial safety and quality assurance framework (SQA) [14].

Cut fruit and vegetables are classified as potentially hazardous foods because they have water activity (a_w_) values above 0.80 and 0.98 and pH values between 3.0 and 7.5, which can be a suitable environment for the development of undesirable microorganisms. Fungi are the main microbial spoilage organisms in fruits and vegetables (*Botrytis* spp., *Alternaria* spp., *Penicillium* spp., *Rhizopus* spp, *Colletotrichum* spp., *Monilinia* spp. and *Sclerotinia* spp.) and bacteria (*Listeria monocytogenes*, *Escherichia coli, Enterococcus faecalis, Salmonella enterica, Pseudomonas* spp. and *Erwinia* spp.) [15,16,17]. However, the main viruses found in fresh produce known to cause foodborne diseases are Norovirus and hepatitis. A virus study conducted in the USA from 1998 to 2007 showed that the microbial contamination of fresh produce caused approximately 18% of all foodborne illnesses, and more recently, a study carried out from 2004 to 2013 revealed that approximately 24% of all foodborne illnesses were caused by contaminated fresh produce (CSPI, 2015). As fresh produce is normally consumed raw or with minimum handling, it is essential to reduce the microbial load of fresh produce as much as possible in order to prevent foodborne illnesses [18].

As previously mentioned, the preliminary treatments the raw materials undergo (peeling, slicing, dicing, shredding, etc.) cause mechanical and physiological damage to the produce leading to a series of chemical and enzymatic stress reactions induced by oxidative processes. As a result of these reactions, undesirable phenomena occur in the exposed wound areas, such as loss of tissue firmness, enzymatic browning and primarily microbial infection facilitated by the initiation of oxidation [19,20,21]. In order to extend the shelf life of fresh produce, ensure produce safety and increase its commercial value, it is essential to monitor the cold chain conditions throughout all stages of produce processing. The quality of the raw materials is of paramount importance as only choice vegetables should be used for fresh-cut processing.

Cutting and washing operations are the most important steps in the RTE fresh produce preparation process. As already reported, when cutting fresh produce, it is important to limit the effects of wounding stress in order to reduce oxidation phenomena, enzymatic activities and microbial development [21]. Using sharp rotary cutters can minimize fruit and vegetable tissue damage. Washing fresh produce in water is normally carried out to promote the cicatrization of cutting wounds and slow down the physiological processes of produce senescence. Moreover, industrial washing systems add sanitizers to the water in order to reduce the microbial load present on the surface of the produce (Figure 1). Chlorine is the most commonly used sanitizer, which is applied at concentrations of 50–200 ppm as hypochlorous acid for 1–3 min [22] or chlorine dioxide (ClO_2_). Chlorine is also applied as hypochlorite (NaOCl), but it has been observed that in the presence of organic matrices, it is likely to form halogenated by-products (trichloromethane) and other by-products that are potentially dangerous to the environment and human health [23]. For this reason, alternative washing treatments have been investigated with the aim of testing more effective processes and avoiding the associated risks.

Other “eco-friendly” sanitizers for disinfecting cut products and preserving their quality and safety include peroxyacetic acid, hydrogen peroxide, weak organic acid (i.e., citric, ascorbic, oxalic and lactic acid), calcium, sodium and potassium derived salts, plant essential oils (such as carvacrol, eugenol and thymol), electrolyzed water and ozone [24,25,26,27,28,29,30]. There are some natural edible coatings that are used to protect the exposed surfaces of fresh-cut produce that can be applied using different techniques including dipping, spraying and brushing. These coatings provide fresh-cut produce protection against gases, moisture loss and microbial contamination: thus, they provide an extension of shelf life. The coatings are made from proteins (soybeans, milk, corn, wheat, peanut), polysaccharides (cellulose, chitosan, alginate, pectin, carrageenan, gum Arabic) or lipids (vegetable and mineral oils, paraffin waxes, nano and micro emulsions) that are applied to produce surfaces in singular or in composite solutions [31,32,33].

### 1.3. Technologies for Quality Preservation

While whole fruits and vegetables have a longer shelf life, fresh-cut produce can only be stored at temperatures below 8 °C for a short time (EUR-LEX Document 32007R1580). The commercial shelf life of sliced vegetables is a few days (on average 4–7 days), while for sliced fruit, it is a little longer (7–10 days). The conventional technologies used to preserve fresh-cut products and maintain the stability of the organoleptic properties and product quality are based on refrigeration in cold storage rooms [34,35]. However, chilling injury may occur when different varieties of fresh-cut fruits and vegetables are kept under the same storage temperature. Refrigeration combined with modified atmosphere packaging technologies (MAP) are often used: the fresh-cut produce is placed in appropriate packaging where the air is removed and replaced by a mixture of gases, thus creating a suitable modified atmosphere for extending shelf life [36,37]. The system requires changes in the O_2_, CO_2_ and N_2_, and the concentration depends on various factors, mainly on the type of produce. The different MAP component gas concentrations used for packaging vary according to the physiological properties of the fresh-cut produce (respiration, ethylene rate), the storage temperature adopted and the gas and water vapor permeability values of the films employed for packaging [24,36,38,39].

Today’s consumers expect healthy produce, ideally free of chemical residues, which has led to the development of innovative technologies, which are mainly used to reduce the microbiological load of minimally processed produce. For example, electron bean irradiation has obtained promising results despite regulatory restrictions (maximum irradiation level for fresh fruits and vegetables to 1.0 kGy, US-FDA), while more recently, ultraviolet C light used in the 200 to 280 nm spectrum and pulsed light (PL) generated by xenon lamps are techniques that have effective germicidal properties, are relatively simple to use and require low-cost equipment [40,41,42,43]. In the last few years, cold plasma and ultrasound techniques have been used that have yet to be evaluated in the scientific papers on the fresh-cut produce. In these cases, researchers show the effects of biological control and promising responses on product quality parameters. [40,44].

### 1.4. Ozone Technology

Post-harvest shelf life and microbial contamination control strategies are often based on the use of chemical compounds that are harmful to human health and the environment. Today, it is of fundamental importance to find a viable alternative to traditional technologies in order to ensure safety and quality [45]. Ozone is one of the most promising treatments for the decontamination of fresh produce with several areas of application in the food industry [46,47].

The chemical reaction required for ozone formation was described by Rice et al. [48]; a diatomic oxygen molecule (O_2_) is divided and each of the two oxygen radicals (O) react with another diatomic oxygen to form the triatomic ozone molecule (O_3_). The rearrangement of atoms to form the ozone molecule requires a large amount of energy that is usually generated by ultraviolet irradiation, electrochemical processes or electrical discharges (electrical corona effect). The first two methods are seldom used due to very high costs and low ozone yields [49].

In 1997, a group of experts from the US Food and Drug Administration (FDA) assigned ozone generally recognized as safe (GRAS) status [50]. In 2001, the FDA approved its use as an antimicrobial agent in the gas and aqueous phases for the treatment, storage and processing of food with the aim of eliminating pathogens and sanitizing food and food production plants [45].

However, to date, the European Food Safety Authority (EFSA) are still validating the use of ozone in the fresh produce industry. The EFSA publication “Scientific Opinions on the risks posed by pathogens in food of non-animal origin (*Salmonella* and *Norovirus* in Leafy greens eaten raw as salads and berries)” suggests using ozone, as well as other sanitizing molecules, for disinfecting leaf vegetables and berries [51].

Ozone technology for extending the shelf life of food is considered a non-thermal method of food preservation that improves food safety without compromising quality and endangering the environment. Since ozone is a highly unstable molecule that auto-decomposes spontaneously and rapidly into oxygen atoms at room temperature, it cannot be collected, stored or transported and must, therefore, be continuously generated in situ [52].

The half-life of ozone (even in the absence of a catalytic destroyer) is very short, usually 30–40 min in water and 2–3 h in air, although these parameters may vary depending on temperature and pH values [53].

The sanitizing and biocidal properties of ozone has attracted the interest of the food and fruit and vegetable sector because the molecule rapidly decomposes into O_2_, without leaving residues on the produce. Although it reacts with some organic compounds present in food matrices, the possible by-products are aldehydes, ketones or carboxylic acids, which do not pose a threat to human health. Contrastingly, other chemicals used as sanitizing compounds for washing fruit and vegetables, such as chlorine (and its derivatives), have proven to produce harmful by-products (chloramines, chlorophenols or trihalomethanes), which trigger cell damage and are strongly associated with the onset of degenerative diseases in humans [54].

Ozone treatments can be applied in both gaseous and aqueous phases. Once produced, ozone can be continuously or intermittently added to the environment in which the produce is stored, or it can be dissolved into water to produce aqueous ozone used for washing and sanitizing; however, it is important to note that it is relatively stable in the gaseous state and highly unstable in aqueous solution [49]. The choice of treatment mainly depends on the type of produce to be treated and the method of application [55].

The equipment costs for the ozone treatment are higher than the costs for the conventional methods of sanitization or fumigation. However, they are largely compensated for long-term applications of the ozone treatments since electrical energy is the main component of cost basis. The power consumption of the ozone generators varies according to the capacity of the oxygen flow in the concentrator and of the gas flow output (it can vary from a few tens of W to a few tens of KW) and is lower in the case of aqueous ozone generators than in gaseous ones [56].

Scientific studies have widely demonstrated the effectiveness of ozone technology, in particular its efficacy in preserving fruits and vegetables [49,53,57,58]. To date, ozone technology has mainly been used in the fish, poultry, dairy (milk and its derivatives) and meat industries, while its use in the fruit and vegetable industry is still limited. Several studies have shown that aqueous ozone is much more effective than gaseous ozone for disinfecting intact products; however, the advantage of gaseous ozone is that it can be used as an antimicrobial agent in liquids. Ozonated water is a good alternative to traditional sterilizing agents because it has proven to be effective in destroying harmful bacteria and viruses even at low concentrations [53]. One of the most powerful natural sanitizers in the world, ozone inactivates up to 99.0% of pesticides and most of the microorganisms present in various food tissues, thanks to its potential oxidizing capacity [59]. Ozone has multiple potential applications as a sanitizing agent in the food industry since it offers significant advantages over traditional antimicrobial agents [55].

Ozone’s sanitizing effectiveness can be influenced by various factors as described in the studies published by Aslam et al. [56] and Tzortzakis and Chrysargyris [45]: type of produce, ozone–produce contact time, storage temperature and relative humidity, pH, microbial load and type of microorganism, location of the microorganisms on fresh produce.

Although studies on the effectiveness of ozone-based treatments have obtained contrasting results, ozone has proven to be an efficient method for fruit and vegetable preservation due to its strong antimicrobial and antioxidant properties. [60,61]. Therefore, ozone treatment is a process that assures food safety and quality [62].

## 2. Literature Review Aims and Objectives

The papers included in the bibliographical review are the studies on aqueous ozone treatments used for washing minimally processed fruits and vegetables, while few studies deal with the use of gaseous ozone in food processing. In this case, ozone treatments are applied to fresh produce cut before or during the packaging process.

Our aim is to discuss the current progress of the research studies on ozone treatments for assuring the quality and microbiological safety of minimally processed fruits and vegetables.

For each scientific study, the tables below show the main qualitative and microbiological characteristics of all fresh-cut produce (firstly for vegetables and secondly for fruits) treated with either aqueous (Table 1 and Table 2) and/or gaseous ozone (Table 3 and Table 4).

## 3. Ozone Treatments and Fresh-Cut Fruits and Vegetables Quality

### 3.1. Physiological and Technological Behaviour

A major challenge for the fresh-cut industry is the cutting process that causes mechanical and physiological damage to plant tissues known as “wound injuries”. As reported in previous studies, these injuries occur by inducing a stress response in the vegetal cells involved in this stress and those nearest the wound, with a physiological stress response that causes an increase in ethylene production (stress ethylene) and the rapid deterioration of the produce. Wounding may cause chlorophyll degradation and yellowing of leafy green vegetables and stems. Moreover, wounding caused by the cutting, peeling and trimming processes increases the respiration rate of fruits and vegetables [105,106]. Wounded plant tissue provides favorable conditions for microorganisms to spoil the produce, which may pose threat to human health. In order to control the negative effects of stress, modified atmosphere packaging has proven to be useful for storing minimally processed fruits and vegetables [107]. In fresh-cut cabbage and apple, the aqueous ozone treatment at 1.4 mg L^−1^ for 1, 5 and 10 min (cabbage) and 1 and 5 min (apple) stimulated the initial respiratory metabolism of cabbage and reduced ethylene production in fresh-cut apple and cabbage, thus enhancing the overall quality of the produce [63,94]. Following gaseous O_3_ treatment with 6.34 mg m^−3^ for 4 h and storing for 12 days at 5 °C in commercial packaging, low respiration (0.61 μLC O_2_ kg^−1^ h^−1^) and ethylene production rates (1.04 μL kg^−1^ h^−1^) were observed in minimally processed melon fruits [103]. The decontamination of fresh spinach with ozonated water at 3 ppm for 4 min at 4 °C inhibited ethylene production for up to 7 days of storage compared to peroxyacetic acid treatments (Tsunami 100^TM^ solution at 300 ppm for 4 min). No significant differences between the two treatments regarding the atmosphere created inside the polypropylene packages were found, and the barrier layer did not affect the respiration rate pattern [27]. Chauhan et al. [80] observed that that the maximum decrease in the respiration and ethylene rates of carrot stick samples stored for 30 days were obtained by combining ozone water treatment (200 mg O_3_ h^−1^) for 10 min followed by cold storage at 6 °C under controlled atmosphere (2%O_2_, 5%CO_2_ and 93%N_2_).

Color is a factor that significantly influences consumer purchase behavior towards RTE fruit and vegetables. In fact, it is usually visual color evaluation that affects consumer perception of freshness, firmness, taste and sweetness as well as the expected degree of ripeness [87]. Moreover, color may play a fundamental role in consumers’ perception of flavor and texture of food. The color of fruits and vegetables depends on their content of natural pigments such as chlorophylls, carotenoids and anthocyanins, as well as other pigments resulting from enzymatic and nonenzymatic reactions [49]. It has also been observed that color variations may be due to the enzymatic browning of tissues by phenolic oxidation over time [108]. Some studies have investigated color changes in the following fresh-cut vegetables treated with ozonated water during shelf life in cold storage: parsley leaves stored for 15 days; cabbage stored for 12 days, lettuce stored for 14 days, rocket leaves stored for 12 days, broccoli florets stored for 6 days and spinach stored for 12 days. The results revealed that ozone treatments did not affect color parameters or the most relevant color pigments (chlorophylls a and b, carotenoids) [27,64,65,69,72,97]. In red peppers, color was not affected by O_3_ immediately following treatment (0.7 ppm for 1, 3 and 5 min), and no surface discoloration was observed during storage at 10 °C for 14 days [81]. Furthermore, no significant differences were observed in color parameters (L*—lightness, a*, b*—chromaticity coordinates, and hue angle—color tone, defined by the International Commission on Illumination—CIE, 1976) in fresh-cut broccoli treated with 2 μL L^−1^ ozonated water and stored at 5 °C for 9 days [82]. Fresh-cut paprika pieces were washed for both 90 and 180 s with various rinsing solutions: tap water, chlorinated water (100 mg L^−1^), electrolyzed water and ozonized water (4 mg L^−1^). They were then stored in polypropylene bags at 5 °C for 12 days, and color was evaluated by comparing the hue angles of the various treatments during storage. The study revealed that long ozone contact times resulted in slightly lower hue angle values than short contact time, which is a positive effect [79].

The main cause of detrimental changes in the texture of fresh-cut produce is physiological cell wall degradation that occurs during and after ripening, which in cut fruit is accentuated by the wounding stress caused by cutting and by metabolic phenomena triggered by the greater freshly exposed surface of plant tissue. Pectinolytic enzymes are mainly responsible for fruit softening as they act on the lamella mediana component, which rapidly leads to increases in porosity and intercellular spaces. Among these enzymes, pectin methylesterases (PMEs) and polygalactorunases (PGs) are those mainly responsible for the loss of firmness. PMEs and PGs act sequentially: PMEs demethylate pectins, while PGs act on these demethylated polymers to break glucosidic bonds and degrade the cell wall [109,110]. In a study by Toti et al. [4], it was observed that gaseous O_3_ treatment reduced the activities of the cell wall enzymes α-arabinopyranosidase, β-galactopyranosidase and polygalacturonase, which delayed fruit softening and maintained the quality of cantaloupe melon. The main issues encountered in fresh-cut processing are dehydration and tissue shrinkage, which can lead to quality defects such as texture alterations. Texture is a critical quality parameter, and loss of firmness may be due to tissue degradation [111].

No significant changes were observed in the texture and moisture content of fresh-cut “iceberg” lettuce samples washed in chlorinated water (100 mg L^−1^), organic acids (5 g L^−1^ citric acid and 5 mL L^−1^ lactic acid) and ozonated water (4 mg L^−1^) during storage for 12 days [85]. Contrastingly, Rico et al. [112] compared three treatments applied to fresh-cut lettuce samples—1 mg L^−1^ ozone at 18–20 °C, 15 g L^−1^ calcium lactate at 50 °C and a combination thereof—which were kept in cold storage for 10 days. It was observed that the ozone treatment exhibited a negative effect on textural properties, since the decrease in texture-related PME activity was correlated with a lower crispness coefficient. It is possible that the controlled activation of PME may improve texture, as it increases cross-linking between pectin chains and cations [113]. In another study by Ummat et al. [68] on capsicum shreds (fresh green bell peppers cv. “Bachata”) treated with ozone at 1, 1.4, 2, 2.4 and 3 mg L^−1^, respectively, for 1, 3 and 5 min and then stored under refrigerated conditions at 5 °C in polypropylene packages, it was observed that ozone treatments had no negative effect on the firmness of the pepper fruits during storage. These findings are consistent with those obtained by Botondi et al. [26], Alexandre et al. [58] and Aguayo et al. [101] who reported that ozone treatment did not affect the texture of fresh-cut muskmelon fruits, green bell peppers and “Thomas” tomatoes.

### 3.2. Chemical and Nutritional Content

The use of ozone in the food industry is limited to food surface hygiene only, and it does not penetrate deeply into the produce. The surface penetration depth for aqueous ozone treatments was limited to few millimeters due to the significant time required for diffusion, while gaseous ozone was able to penetrate much deeper (approximately 10 cm) within a few minutes [114].

Maintaining the quality of fruit and vegetables depends on the ability of the plant tissues to preserve their nutritional and safety characteristics for a long time. As previously mentioned, minimally processed produce deteriorates rapidly due to the damage triggered by cutting processes, which cause metabolic stress (especially oxidative stress) leading to reduced shelf life and quality loss. Therefore, any negative impact on nutrient content caused by ozone can be assumed to be limited to surface only [56].

#### 3.2.1. Total Soluble Solids and Titratable Acidity

In general, the total soluble solid (TSS) content of fruit is an important quality parameter, in close parallel with the content of organic acids (titratable acidity). Increased TSS values and reduced titratable acidity during the shelf life of minimally processed products depend mainly on the water loss and hydrolysis of polysaccharides such as starch to form soluble sugars and increased respiration of organic acids [115]. Regardless of the type of produce and of the ozone treatments used, in the studies examining fresh-cut tomatoes [74], melon fruits [26] and pineapple [95], this did not significantly change the trends of ozonated samples compared to untreated produce. Recently, Ummat et al. [68] showed that irrespective of the treatments applied to green bell peppers (from 1 to 3 mg L^−1^), the TSS contents proved to be similar amongst ozone-treated samples and control samples during storage. However, Amaral et al. [116] noticed a different trend in the TSS profile of fresh-cut green bell peppers compared to untreated samples. The produce was washed with tap water and two concentrations of ozone for 1 min, dried and immediately analyzed. It was observed that the peppers sanitized with tap water and 1.8 mg L^−1^ of ozonated water were statistically equal with Brix values of 4.25°, while a different value (4.0°Brix) was obtained for the samples treated with ozonated water at 1.6 mg L^−1^.

#### 3.2.2. Fresh-Cut Browning

When fresh-cut produce is cut and exposed to air oxidization occurs, depending on its structural and molecular composition, since there is a considerable increase in enzyme activity on the surface of fresh produce. As a result, dark pigments generated by the oxidation of polyphenols appear, which are the products of reactions catalyzed by the enzyme phenylalanine ammonium lyase (PAL). PAL activity is closely related to browning. Browning occurs when injured plant tissue (caused by cutting and slicing procedures) comes into contact with oxygen and a key enzyme named polyphenol oxidase (PPO) changes phenolic compounds into quinones through oxidation, which then polymerize to form dark pigments [117].

Another enzyme that acts as a “scavenger” of the free radicals formed in this process is peroxidase (POD). The intensity of browning phenomena is influenced by the level of enzyme activity and by the polyphenol content of the tissue.

As ozone is a strong oxidizing agent, it tends to react with the components in contact with the cut tissue, thus creating high reactive species referred to ROS such as OH∙, ∙HO_2_, O_2_− and O_3_− ions, which leads to increased degradation. Ozone also binds to the double bonds of molecules such as carotenoids, chlorophylls and anthocyanins and oxidizes them, which can result in changes in tissue color [118]. This degradation seems to be strongly dependent on the ozone concentrations used and/or on the treatment times.

Previous studies have revealed that ozone has an inhibitory effect on the enzymes responsible for browning in fresh-cut celery and lettuce, probably due to ozone’s high oxidation potential [89,90,112]. In order to corroborate the latest data, Pongprasert et al. [71] conducted a study on a new ozone micro-bubble technique for controlling microbial growth and browning in fresh-cut lettuce. The authors confirmed that disinfection by ozone microbubbles at a concentration of 5 ppm for 5 min inhibited microbial growth and PPO activity for up to six days of storage at 4 °C, which is correlated with lower amounts of quinone during storage. The authors concluded that the inhibitory effect of the browning enzyme and substrate resulted in less browning of fresh-cut lettuce during storage. Another study [78] conducted on sliced potatoes investigated the effectiveness of acidulant dip treatments (SAS and NatureSeal PS-10 commercial dip provided from NatureSeal^®^-ReduSal, Mantrose-Haeuser Co., Inc., Westport, CT, USA) with or without an additional 2 ppm aqueous ozone dip treatment for 2 min. The results obtained during cold storage at 4 °C for 28 days showed that the ozone-treated samples of fresh-cut potatoes showed reduced enzymatic browning than the samples that had not undergone ozone dip treatment. However, ozone did not appear to have any effect on aerobic plate counts or PPO activity. In contrast to previous studies on potato slices and fresh-cut lettuce, it was observed that ozone has no anti-browning properties. After 14 days of storage at 4 °C under MAP conditions, fresh-cut potato strips dipped in ozonated water or ozone–Tsunami exhibited a significant increase in browning due to prolonged storage [91]. Similarly, in the case of fresh-cut lettuce washed in ozonated water (3, 5 and 10 ppm) for 5 min at room temperature and stored at 10 °C, the different ozone concentrations did not affect PAL activity. Browning, which is measured as a* value, increased depending on the ozone concentrations adopted, while the combined treatment of hot water at 50 °C for 2.5 min followed by ozonated water (5 ppm for 2.5 min) significantly inhibited PAL activity for up to 3 days of storage. This combined treatment curbed increases in the a* value, thereby delaying the browning effect for up to 6 days of storage [88].

#### 3.2.3. Antioxidants

Antioxidants are important compounds commonly found in fruit and vegetables. These molecules play an important role in defending plant tissue against oxidative stress by binding to free radicals and preventing damage. Antioxidant defense systems generally include both enzymatic and non-enzymatic systems. The main non-enzymatic antioxidants include ascorbic acid, polyphenols, carotenoids and anthocyanins, while enzymatic antioxidants include superoxide dismutase (SOD), peroxidases (in the ascorbate-APX, glutathione-GPX forms) and catalase. All of these antioxidants enable plant cells to scavenge ROS and help to prevent cellular damage [119]. In this regard, Chen et al. [98] investigated the metabolic activity of some antioxidant enzymes found in fresh-cut green peppers including POD, SOD and PAL activities (that were induced by ozone supplied at 6.42 mg cm^−3^ for 15 min and stored under MAP conditions at 5 °C for 21 days) and PPO activity (inhibited by ozone and MAP treatment). These authors concluded that both ozone and MAP can enhance the effectiveness of the antioxidant defense system. The same trend was observed in fresh-cut carrots treated with ozone and stored in controlled atmosphere for 30 days. Reductions in ascorbic acid, carotenoids and oxidative enzymes such as PPO and POD were observed in the samples. In this case, the reduction in POD activity by ozonation implies detoxification, which may be linked to the surface decontamination process, and the reduction in PPO activity could be associated with lower respiration rates detected in the experiment [80].

Ozone can increase plant stress tolerance by stimulating the ROS scavenging system in cells, which enhances the synthesis of the antioxidant enzymes. It is assumed that ozone induces loss of antioxidant compounds due to its strong oxidizing ability. However, ozone oxidative stress may induce defense mechanisms in plant tissue. The sensitivity of fresh produce to ozone varies according to the type and species of vegetable crop [100].

Ascorbic acid and phenolic compounds are undoubtedly the most important antioxidants found in fresh-cut fruit and vegetables. Numerous studies have reported that the total phenolic and ascorbic acid content during storage of different minimally processed produce treated with different concentrations of gaseous ozone and ozonated water tend to decrease compared to their respective controls ([64,99] and 2014 in fresh parsley; [80] in carrot sticks). Reduced levels of ascorbic acid were mainly observed in lettuce [90] after washing in ozonated water (10 and 20 mg L^−1^). The suggested mechanism of action indicates that ascorbic acid can act either directly by oxidizing molecules or through the formation of free radical intermediates [99]. The oxidation of ascorbic acid produces dehydroascorbic acid, which plays an important physiological role. The enzyme responsible for this reaction is ascorbate oxidase (APX), which is mainly activated in response to stress. The activation of APX may, therefore, lead to a decrease in ascorbic acid [120].

However, ozone can also have a positive effect on ascorbic acid levels, which can be explained by preserving its content caused by stimulating the natural plant defense mechanisms through the stress generated by the oxidizing molecule and by increased synthesis and accumulation of antioxidant compounds such as ascorbic acid in the cells. Ummat et al. [68] proved that aqueous ozone treatments above 2.4 mg L^−1^ of minimally processed bell pepper for 14 days significantly reduced the microbial load and showed better retention of ascorbic acid, firmness and color. Similar trends were observed for fresh-cut pineapple fruits treated with aqueous ozone at 0.6, 0.9 and 1.5 ppm and stored for 20 days at 2 °C [95] and for fresh-cut parsley in a study by Zhang et al. [89] who noted that lower ozone concentrations resulted in higher ascorbic acid retention. Other studies have found that the ascorbic acid content of ozonated water-treated samples remained almost constant during storage compared to control samples. This was observed by Olmez and Akbas. [83] and Hassenberg et al. [86] in fresh-cut lettuce treated with 0.5 to 4.5 ppm and with 3.6 ppm of ozone, respectively.

A study on fresh-cut papaya treated with ozone at 9.2 μL L^−1^ for 10, 20 and 30 min to investigate the effect of ozone on antioxidants following a 20 min ozone treatment showed a 10.3% increase in total phenolic content and a 2.3% decrease in ascorbic acid content. Contrastingly, a 30 min ozone treatment caused a decrease in total phenol content [102]. The increase in phenolic content may be due to PAL activity stimulated by different abiotic stresses. Similar results were obtained by Alothman et al. [96] who found that the phenolic content of fresh-cut pineapple and banana increased significantly after exposure to 0.72 mmol of ozone for 20 min. The same authors also indicated a possible mechanism of phenol metabolism involving a sequence of a primary protective anti-oxidative effects followed by the subsequent auto decomposition of ozone and production of free radicals, the latter also determined by a prolonged ozone treatment duration. In fresh-cut cabbage Nie et al. [65] showed the positive effect of aqueous ozone combined with sodium metasilicate on ascorbic acid total phenolic and carotenoid contents, while in a study on the quality (polyphenol content, color and sensorial properties) of fresh-cut lettuce, Wang et al. [67] obtained similar positive results using another combined treatment (ozone and lactic acid). In minimally processed grape berries (cv Thompson Seedless and cv Black) sanitized with ozone at 2, 4, 6, 8 mg L^−1^ and with NaOCl (100 mg L^−1^) treatment and then stored for 21 days at 5 °C, total polyphenol content proved to be 23–50% higher in Thompson Seedless and 18.5–28% higher in Black seedless grapes washed in ozonated water (especially at concentrations of 6 and 8 mg L^−1^) compared to the NaOCl samples [93]. However, contrasting results have been obtained by other authors who found that ozone treatments caused reductions in phenolic contents, as demonstrated in the studies on fresh-cut parsley leaves [64], lettuce [71], apple [94], spinach [27] and parsley treated with gaseous ozone at 950 µL L^−1^ [99].

### 3.3. Sensorial Properties

Consumer acceptability depends on several critical quality parameters described by four different sensorial attributes: attractive color (appearance), acceptable aroma and taste (flavor), appropriate texture [121]. External appearance of minimally processed fruits and vegetables is the main quality attribute affecting consumer choice and determines whether a product is accepted or rejected [75]. Nevertheless, the repeated purchase of the product is driven by expected quality factors such as flavor compounds and texture [73]. Aroma compounds are perceived retro nasally by olfactory receptors in the nose, while taste is the detection of non-volatile compounds by several types of receptors in the tongue [75]. Textural attribute is essential for determining the consumer acceptability: the presence of defects may cause rejection of a fresh produce [122].

Several studies demonstrated that ozone treatment is an alternative method to extend the shelf life of fruit and vegetables, maintaining the sensory quality during storage. Olmez and Akbas [83] evaluated the effects of ozone concentration (0.5–4.5 ppm) and exposure time (0.5–3.5 min) on the overall visual quality, cut edge tissue browning, firmness and aroma of green leaf lettuce. Ozone-treated samples at 2 ppm for 2 min had a better score up to 9 days of storage in all sensory parameters than control and chlorine- and organic-acid-treated samples. Therefore, aqueous ozone washing was the most effective treatment that features slower degradation with the consequent extension of shelf life, maintaining sensorial quality of fresh-cut lettuce.

Another study on the effects of different concentrations of ozone water treatment (10 and 20 mg L^−1^ min and 10 mg L^−1^ min activated by ultraviolet C) on fresh-cut lettuce showed that ozone samples maintained an excellent visual quality, including the appearance features of gloss, freshness and color uniformity and intensity for up to 13 days. At the end of the storage, the fresh-cut produce provided a moderate crispy texture and a typical aroma [90].

In a study conducted by Sothornvit et al. [123] a high score of the overall acceptance of fresh-cut cauliflower treated with ozonated water at 3.1 and 3.5 ppm for 15 min is achieved. The sensory attributes assessed (color, off-flavor and visual quality) were considered acceptable by the consumer up to 18 days of storage. On the other hand, ozone washing method did not result in any significant difference on all sensory attributes of fresh-cut basil.

Nie et al. [65] focused on the effects of aqueous ozone treatment (2 ppm) combined with 0.4% sodium metasilicate acid for 2 min on visual appearance and off-odor of fresh-cut cabbage. This combination treatment achieved acceptable sensorial quality, extending the shelf life of fresh-cut produce up to 12 days.

In another study conducted on fresh-cut melon cubes, gaseous ozone treatment at 10,000 ppm for 30 min maintained the full typical aroma, colour and a very firm texture during the storage. [104].

The following results on fresh-cut melon treated with gaseous ozone at 6.34 mg m^−3^ for 3 h on visual quality, aroma and firmness were obtained. It was observed that gaseous ozone treatment prevented the translucency. This result traced back to a slowing down of the ripening process of the melon slices during the storage and leads to a marketability of the product up to 12 days [103].

## 4. Microbiological Control in Fresh-Cut Fruits and Vegetables

The development and growth of microbial population in fruits and vegetables can occur at different stages of the farm to fork supply chain. Propagation of microorganisms occurs during cultivation in the field, harvesting, post-harvest handling and transportation, storage and processing and marketing for human consumption [124].

Fresh-cut products generally have a_w_ values above 0.80–0.95, and a pH range between 3.0 and 6.5. These conditions allow the development of undesirable microorganisms (spoilage and pathogens). *Erwinia, Pseudomonas, Xanthomonas* and *Pectobacterium* genera are the main spoilage microorganisms that cause qualitative alterations and off-odors, by altering the chemical composition and physical structure of plant products [125]. The main pathogenic microorganisms are toxin-producing *E. coli, Salmonella* spp., *Yersinia enterocolytic, Shigella* spp., *L. monocytogenes, Aeromonas hydrophila* and *Pseudomonas aeruginosa* [126]. Moreover, fresh-cut produce may also serve as a vehicle for viral agents such as *Norovirus* and *Hepatitis A Virus (HAV), Rotavirus* and *Astrovirus* [127]

Microbiological limits are regulated by ISO standards and the normative reference in Europe is the Regulation (CE) 2073/2005 of the European Commission, issued on 15 November 2005 related to microbiological criteria applicable to food products.

Fresh-cut processing of fruit and vegetables can induce physiological, biochemical and morphological alterations in plant tissues, especially during cutting operations, which damage the plant tissue and promote the entry of microorganisms, thus increasing the risk of microbial contamination. Moreover, cutting the fruits allows the leakage of plant cellular fluids that provide a nutrient medium for bacterial survival and growth [125].

The shelf life of fresh-cut fruit and vegetables is quite short, approximately 4–10 days, which depends mainly on the type of produce. There are several techniques that can effectively control the development of microorganisms in minimally processed fruit and vegetable produce: refrigerated storage; acidic solutions that are mainly used to control the growth of *L. monocytogenes* [128]; antimicrobial substances such as garlic and onion extracts [129]; bio preservation, a technique of extending the shelf life of foods by using useful microorganisms that produce natural antimicrobial compounds such as bacteriocins produced by *Leuconostoc* spp. against *Listeria* spp. [130,131].

Ozone technology is an innovative sanitation technique used for the decontamination of fresh-cut produce due to its antimicrobial action. Ozone effectively controls microbial growth due to its high reactivity and strong oxidizing potential and is a more efficient biocide than other chemical substances used [132]. Several studies have shown that low ozone concentrations can inactivate a broad spectrum of microorganisms; however, when combined with other sanitizing techniques, such as UV radiation, hydrogen peroxide and hydrostatic pressure, the microbial killing capacity of aqueous and gaseous ozone is even more effective [90,91,104,133,134].

Micro-nano-bubbles (MNBs) are an innovative method for prolonging the reactivity of aqueous-phase ozone and enable us to reduce the growth of bacteria, total yeasts and molds in food and food processing environments with the aim of maintaining freshness and quality and extending the shelf life of fresh-cut produce [135].

### 4.1. Aqueous Ozone Treatments

The efficiency of aqueous ozone depends on the different ozone concentrations, temperature and mainly the contact time between the ozonated water and the produce during the washing process.

Immersing fresh-cut lettuce (*Lactuca sativa*) and bell peppers (*Capsicum annuum)* in pre-saturated ozonated water (0.5 mg L^−1^) resulted in a lower microbial reduction compared to immersion in continuously ozonated water (0.5 mg L^−1^), which leads to a reduction of 2 log after 15 min and 3.5 log after 30 min of exposure [76].

Renumarn et al. [77] analyzed fresh-cut broccoli (*Brassica oleraceae* L.) treated with aqueous ozone (2.5 mg h^−1^) for 5, 10 and 15 min. The most effective ozone concentrations used were 0.56, 1.00 and 1.50 ppm. The results showed reductions in total coliform bacteria, molds and yeasts of 1.20, 2.50 and 1.80 log_10_ CFUg^1^, respectively, compared to the control treated with pure water. Several studies have reported the influence of ozonated water (1.4 mg L^−1^) for different treatment times on fresh-cut apple slices (5 and 10 min) [94], onion (1, 3 and 5 min) [66] and cabbage (1, 5 and 10 min) [63]. All ozone treatments showed that longer time of aqueous ozone treatments was more effective in decreasing the final total bacteria, mold and yeast counts during storage, compared to control samples.

The results of the study conducted by Nie et al. [65] showed that the combined action of ozonated water at 2 ppm and sodium metasilicate at 0.4% for 2 min reduces *E. coli* O157:H7 counts by 1.78 log_10_ CFUg^1^, compared to the control sample, and inhibits the growth of bacteria, yeast and mold after 12 days of storage.

However, ozone combined with other antimicrobial treatments did not always obtain positive results; for example, fresh-cut lettuce treated with 1% lactic acid for 90 s, followed by aqueous ozone treatment of 1 mg L^−1^ for 30 s, did not reduce the bacterial count of *Escherichia coli* after 5 days of storage at 5 °C. Bacterial colonies were counted after 0, 3 and 5 days, and the average logarithmic reduction in microbial load was approximately 0.8 log CFUg^−1^ [67].

Several studies have demonstrated that treatment temperature is another important parameter to consider. Fresh-cut lettuce treated with 2 mg L^−1^ ozonated water was tested at two different temperatures, 4 and 15 °C. The treatment carried out at 4 °C significantly reduced the microbial load of *Salmonella enterica* serovar Typhimurium and *Escherichia coli* inoculated onto the surface of the lettuce leaves without altering their visual and sensory properties [69].

In another study, low temperatures (5 °C) were used to wash samples of tomato slices with ozonated water (0.4 mg L^−1^) for 1, 3 and 5 min, which were then stored for 14 days and destined for the fresh-cut industry. The 3 min contact time proved to be the most effective for reducing total bacterial count values: the mesophilic, psychrotrophic and yeasts’ load was reduced by at least 1CFU g^−1^, compared to the control sample. In this case, mold growth remained the same showing values of approximately 2.1CFU g^−1^ for all samples. Long treatment times do not improve microbiological safety, probably due to the presence of organic compounds with high ozone demand on fruits and vegetables surface that may inactivate the ozone before it can reach microorganisms [74]. The 3 min treatment time proved to be more effective in the study conducted by Alexandre et al. [58], who, after being washed in ozonated water (0.3 and 2 mg L^−1^) for 1, 2 and 3 min, *Listeria innocua* in fresh-cut peppers, total mesophilic bacteria in fresh-cut strawberries and total coliforms in fresh-cut watercress were tested, resulting in microbial load reduction of 2.8, 2.3 and 1.7 log CFUg^−1^, respectively, compared to the control. A 1.7 microbial log reduction was also observed in fresh-cut celery, treated with ozonated water at 0.18 mg L^−1^ by Zhang et al. [89].

Besides treatment times and temperature, the antimicrobial efficacy of aqueous ozone treatments depends on the ozone concentration used. In a study conducted on fresh-cut spinach, different concentrations of ozone in water were tested, 0.4, 0.8 and 1.2 mg L^−1^, with treatment times for each thesis of 1, 15 or 30 s. Positive results were obtained with 0.8 mg L^−1^ of ozonated water for 30 s, since it inhibited microbial growth and maintained the characteristics of the fresh-cut spinach over the first 5 days of storage [70].

### 4.2. Gaseous Ozone Treatments

Gaseous ozone treatments require longer contact time than ozonated water treatments; in a study conducted on fresh-cut papaya, gaseous ozone treatment for 20 min proved to be more effective for reducing total coliform count (0.39–1.12 log_10_CFUg^−1^) than 10 and 30 min treatments [102]. Moreover, gaseous ozone treatment on previously processed produce has a greater effect on the reduction in final microbial load. In another study, fresh-cut red peppers were treated with 0.7 ppm of gaseous ozone for 1, 3 and 5 min and were then packed in polypropylene bags under MAP conditions and stored at 10 °C for 14 days. The results showed equal initial counts of aerobic mesophilic bacteria for control and for ozone treatments; however, from day 14, a reduction of approximately 2.56 log units was observed [81].

Selma et al. [104] reported using gaseous ozone at 10,000 ppm for 30 min under vacuum for treating fresh-cut cantaloupe in order to determine the load of *Salmonella* in ripe and unripe fruits, showing a reduction of 2.8 and 4.2 log_10_CFU/rind disk (ring disk of 12.6 cm^2^), respectively.

In another experiment, a flow of ozone-enriched air at 4 ± 0.5 μL L^−1^ for 30 min every 3 h was applied to fresh-cut “Thomas” tomatoes, stored at 5 °C for 15 days. Gaseous ozone was more effective at reducing bacterial load (from 1.1 to 1.2 log_10_ units) than fungal count (0.5 log_10_ units). A greater level of ozone antimicrobial effectiveness was observed when produce was exposed to a higher ozone concentration (7 μL L^−1^), which extended produce shelf life, and the organoleptic and biochemical characteristics of the produce remained unchanged [101].

Gaseous ozone treatments on food products can be combined with other sanitizing technologies to enhance disinfection potential. Ozone treatments combined with UV-C radiation were applied to fresh-cut rocket for 12 days at 5 °C in order to monitor qualitative, sensory and microbiological parameters. The synergistic antimicrobial activity of these technologies does not affect the sensory quality of the produce, which remained unchanged throughout the storage period; however, it induced a reduction in microbial load from storage day 8, compared to gaseous ozone and chlorine treatments [97].

## 5. Conclusions

There is an increased demand for RTE products, particularly fresh-cut fruits and vegetables, especially in the most industrialized economies, which is mainly due to the availability of technologies that preserve the quality and safety of fresh produce and ensure maximum shelf life.

Several studies have investigated the sanitation potential of ozone for processing fresh-cut produce with the aim of preserving the nutritional and sensory quality characteristics and obtaining microbiological control to stop or slow down food spoilage. Sanitizing treatments reassure consumers that the treated produce is hygienically clean and safe.

Although in some studies the results were conflicting, in accordance with several papers included in this review, ozone treatments applied at appropriate concentrations and contact times (in aqueous and/or gaseous form and/or in combination with other compounds/technologies), adjusted according to the different RTE produce and combined with good agricultural practices, GMP and HACCP methods appear to provide promising qualitative and biological results.

## Figures and Tables

**Figure 1 foods-10-00748-f001:**
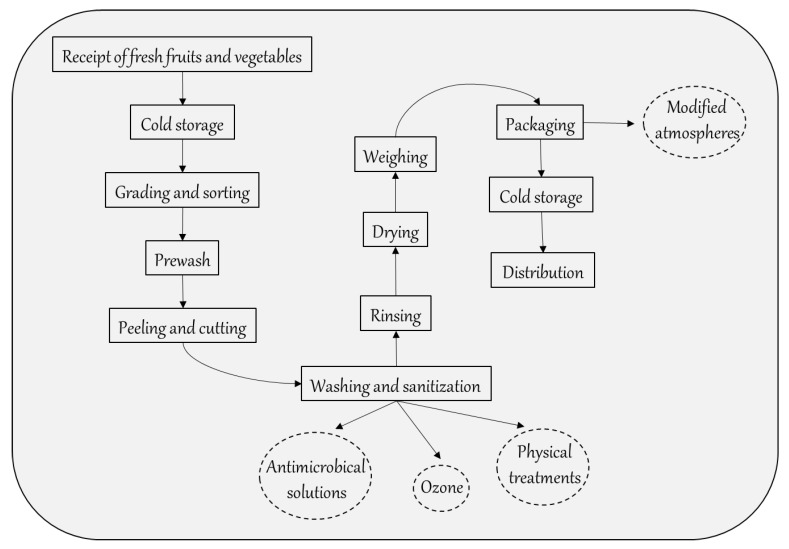
Flow sheet of cut fruit and vegetable products.

**Table 1 foods-10-00748-t001:** Effect on the quality parameters and microbiological control of fresh-cut vegetables treated with aqueous ozone.

Vegetables	Treatment	Qualitative Results	Microbiological Results	References
Cabbage	Aqueous ozone (1.4 mg L^−1^) for 1, 5 and 10 min	The aqueous ozone (AO) treatment stimulated initial respiratory metabolism, reducing ethylene production and improved the overall quality of fresh-cut cabbage, compared with that of the control. The effect of treatment for 5 min on the removal of trichlorfon, chlorpyrifos, methomyl, dichlorvos and omethoate was greater than that of the control.	The growth rates of aerobic bacteria, coliforms and yeasts were significantly inhibited by ozonated water; longer treatment time (10 min) showed the greatest inactivation of bacteria, coliforms and molds.	Liu et al., 2021 [63]
Parsley leaves	Aqueous ozone 12.0 mg L^−1^ compared with chlorine 100 mg L^−1^ (5 min)	Chlorophylls, ascorbic acid, total phenolic contents and antioxidant activity were not adversely affected by washing treatments tested (aqueous ozone and chlorine).	Reduction in *E. coli* counts were observed in the samples, while *L. innocua* counts were stable during storage. Maximum reductions in the counts of *L.innocua* and *E.coli* were obtained with chlorinated water treatment. However, reduction in *L. innocua* count obtained with chlorinated water was not significantly higher than that obtained with ozonated water. Therefore, ozone could be considered as an alternative to chlorine for *L. innocua*	Karaca and Velioglu, 2020 [64]
Cabbage	Aqueous ozone 2 ppm combined with 0.4% sodium metasilicate for 2 min	This combination treatment had no negative effects on sensory characteristics (appearance and off-odor), color and the contents of ascorbic acid, total phenols and carotenoids.	This combination treatment achieved acceptable microbial quality and reduced *E. coli* counts by 3.33 logs units compared with the control samples after 12 d of storage.	Nie et al., 2020 [65]
Onion	Aqueous ozone 1.4 mg L^−1^ for 1, 3 and 5 min	AO treatment for 1 min significantly reduced the weight loss of fresh-cut onions during a longer storage time (8–14 days). All AO treatments reduced the respiration rate and the softening of fresh-cut onions. AO treatment for 5 min significantly reduced the residual levels of five tested pesticides (dimethyl dichlorovinyl phosphate, cypermethrin, chlorpyrifos, methomyl and omethoate) compared with water treatment.	The results showed an inhibition of the growth of aerobic bacteria, coliforms and yeasts during storage, with the AO treatment for 5 min allowing the lowest growth rates.	Chen et al., 2020 [66]
Lettuce	Aqueous ozone 1 mg L^−1^ (30 s) or 2 mg L^−1^ (30 s) with 1% lactic acid (90 s)	Quality analysis (color, sensory qualities, electrolyte leakage, polyphenolic content and weight loss) showed that lactic acid (LA) + AO did not cause additional quality loss compared with tap water treatment.	Microbial analysis showed that LA plus AO led to the greatest reductions in microbes (*Escherichia coli* O157:H7, aerobic mesophilic counts, aerobic psychrophilic counts, molds and yeasts) during storage (0–5 days at 5 °C)	Wang et al., 2019 [67]
Green bell pepper	Aqueous ozone 1–3 mg L^−1^ 1–5 min	The exposure to ozone treatments above 2.4 mg L^−1^ for higher durations showed better retention of other quality parameters such as ascorbic acid, firmness, color and overall acceptability.	The exposure to ozone treatments above 2.4 mg L^−1^ for higher durations significantly reduced the microbial load.	Ummat et al., 2018 [68]
Lettuce	Aqueous ozone 2 mg L^−1^ at different temperatures (4 and 15 °C)	During storage period at 4 °C for 14 days, the highest quality was observed from the samples treated with cold ozonated water. Any effect on color properties.	Cold ozone treatment (4 °C) significantly reduced the natural background microflora of lettuce. *Salmonella enterica serovar Typhimurium* and *Escherichia coli* inoculated on lettuce samples were insignificantly influenced by the temperature of water.	Sengun et al., 2018 [69]
Spinach	Aqueous ozone 0.4, 0.8 and 1.2 mg L^−1^ for 1, 15 or 30 s	Ozonated water (0.8 mg L^−1^ for 30 s) before packaging reduced yellowing and maintained compositional characteristics of the fresh-cut spinach leaves, ensuring a shelf life extension of 3 days.	The microbiological analyses indicated the ability of ozonated water to decrease the Gram-negative and Enterobacteriaceae sp. load only during the first 5 days of storage (0.8 mg L^−1^ for 30 s).	Papachristodoulou et al., 2018 [70]
Lettuce	Aqueous ozone 0.5 ppm for 5 min	The amount of phenolic compounds of the lettuce was reduced by AO. Washing with AO also inhibited polyphenol oxidase (PPO) activity for up to six days of storage at 4 °C, which is correlated with a lower amount of quinone content during storage. An inhibitory effect of browning enzyme and substrate resulted in a lower browning symptom.	For total bacteria counts, coliform counts and yeast and mold, ozone microbubbles (O_3_−MBs) treatment resulted into a 1–2 log reduction, which was similar to the result achieved by 100 mg L^−1^ chlorinated water.	Pongprasert et al., 2016 [71]
Broccoli	Aqueous ozone 0.56 ppm ozone for 5 min and 1.60 ppm for 3 min)	This treatment did not have negative effects on color (lightness, a*, b* and hue angle values), chlorophyll content or sensory attributes (overall visual quality, visible color and odor).	All treatments reduced the amount of microbes compared with the initial microbial loads of unwashed fresh-cut broccoli. Treatment with 1.60 ppm was the most effective treatment with regard to reducing aerobic bacteria, coliforms and yeasts and molds.	Renumarn et al., 2014 [72]
Lettuce, spinach and parsley	Aqueous ozone 12 mg L^−1^ Gaseous ozone 950 μL L^−1^, 20 min	AO does not affect chemical characteristics of the vegetables (chlorophyll a, chlorophyll b, ascorbic acid and total phenolic contents and antioxidant activity). Gaseous ozone (GO) caused significant losses in important bioactive compounds of parsley. Ascorbic acid and total phenolic contents and antioxidant activity in ozone-treated samples were 40.1, 14.4 and 41.0%, respectively, less than the control samples.	Aqueous ozone: chlorine and ozone washes resulted in average log units reductions of 2.9 and 2.0 for *E. coli* in the vegetables tested, respectively, while the efficiency of ozone (2.2 log) was very close to that of chlorine (2.3 log) on *L. innocua.* GO treatment resulting in 1.0–1.5 log reductions in the numbers of *E. coli* and *L. innocua* in parsley.	Karaca and Velioglu, 2014 [73]
Tomato slices	Aqueous ozone 0.4 mg L^−1^ for 1, 3 and 5 min	Ozonated water treatment of 3 min achieved the best firmness retention and reduced the consumption of fructose and glucose. The use of ozonated water did not affect the total acidity, pH, total solid soluble, organic acid as ascorbic, fumaric or succinic acid and the sensorial parameters.	Ozonated water treatment of 3 min achieved the best microbial quality (mesophilic, psychrotrophic and yeast load)	Aguayo et al., 2014 [74]
Bell peppers	Aqueous ozone 1 μL L^−1^ Gaseous ozone 0.7 μL L^−1^	In all the experiments, O_2_ continuously decreased and CO_2_ concentration increased. The pH value increased, and a significant softening was observed in all the fruits. By day 14, L values decreased in all the fruits, with the greatest changes found in the chlorinated samples (approximately 12 units). The aqueous solutions, ozonated (1 μL L^−1^) and chlorine (200 μL L^−1^) water showed greater changes in the quality attributes with increasing washing times. The GO treatment had no effect on the physicochemical attributes studied.	The exposure for three min to gaseous ozone reduced the mesophiles, psychrotrophes and fungal populations of the fresh-cut peppers in 2.5, 3.3 and 1.8 log units, respectively, compared to ozone treatment in water.	Horvitz et al., 2014 [75]
Lettuce and green bell pepper	Aqueous ozone 0.5 mg L^−1^		This vegetables dipped in chlorinated water (20 ppm) resulted in a 1 log decrease in the total microbial count in the first 15 min. The immersion of vegetables in water pre-saturated with ozone did not make any difference because the total microbial count decreased approximately 0.5 log for the same time. Sanitation treatments were most effective when vegetables were dipped in continuously ozonated (0.5 mg L^−1^) water, leading to about 2 log of microbial load decrease in the first 15 min and 3.5 log after 30 min of exposure.	Alexopoulos et al., 2013 [76]
Broccoli florets	Aqueous ozone 0.56, 1.00 and 1.50 ppm	Ozonated water treatments impacted fresh-cut broccoli quality by decreasing the chlorophyll contents, L* value and hue angle. The visual quality and visual color evaluated by the sensory panel in both ozonated water and non-ozonated water treatments were not significantly different.	Application of ozonated water for 15 min significantly reduced coliforms, total bacteria and yeast and mold counts by 1.20, 2.50 and 1.80 log_10_ CFU.g^−1^, respectively, when compared with that of the control.	Renumarn et al., 2013 [77]
Spinach	Aqueous ozone 3 ppm for 4 min	The treatments with ozonated water (3 ppm for 4 min) and water tsunami solution (300 ppm for 4 min) allowed the color to be maintained during storage. Moreover, results suggest that sanitization with ozonated water causes an initial dejection of ethylene production in addition to a phenolic content decrease coupled with a lower antioxidant activity.		Bartoloni et al., 2012 [27]
Potatoes slices	Aqueous ozone 2 ppm	Acidulant dip treatments with aqueous ozone (2 ppm) had significantly higher L-values and lower a-values. NatureSeal (NS) and sodium acid sulfate (SAS) were the most effective acidulant treatments in reducing browning (higher L-values, lower a-values and browning index values) regardless of ozone treatment. Ozone did not appear to have any effect on polyphenol oxidase (PPO) activity. NS and SAS also had lower PPO activity compared to other treatments.	Ozone did not appear to have any effect on aerobic plate counts (APCs). NS and SAS also had lower PPO activity compared to other treatments and significantly lower APCs.	Calder et al., 2011 [78]
Paprika	Aqueous ozone 4 mg L^−1^ for 90 and 180 s	All washing solutions (tap water, chlorinated water 100 mg L^−1^ and pH 6.5–7, electrolyzed water pH 7.2 and ozonized water 4 mg L^−1^) showed insignificant differences in gas composition, and no off-odor was detected. Longer contact time resulted in slightly lower hue angle value than a short one for all washing solutions. Samples washed with ozone washings showed lower electrolyte leakage than other washing solutions.	Samples washed for longer contact time except those washed in ozonized water showed increased microbial numbers during storage.	Das and Kim, 2011 [79]
Red bell peppers, strawberries and watercress	Aqueous ozone (0.3 e 2 mg L^−1^) for 1, 2 and 3 min		The highest microbial reductions were obtained for the highest concentration with the highest treatment time (3 min). Listeria innocua in fresh-cut peppers, total mesophilic bacteria in fresh-cut strawberries and total coliforms in fresh-cut watercress were tested, resulting in a microbial load reduction of 2.8, 2.3 and 1.7 log cycles, respectively, compared to the control.	Alexandre et al., 2011 [58]
Carrot sticks	Aqueous ozone 200 mg h^−1^ for 10 min	The maximum decrease in respiration and ethylene emission rates were obtained by the combination of CA with ozone. Significant reduction in ascorbic acid, carotenoids and oxidative enzymes such as polyphenol oxidase (PPO) and peroxidase (POD) were observed due to ozonation and CA storage. The control of lignification by ozone in synergy with CA was characterized by a decrease in L values.	The ozonation in combination with CA have a positive role in controlling microbial spoilage.	Chauhan et al., 2011 [80]
Red pepper	Aqueous ozone 0.7 ppm and chlorinated water (200 ppm) for 1, 3 and 5 min	Weight loss was negligible. O_2_ concentration decreased and CO_2_ levels increased continuously, with no differences between treatments for the ozonated samples. With chlorine, changes in the gas composition were more accentuated. AO treatments does not affect physicochemical parameters. Color was not affected by O_3_ immediately after the treatment, and no surface discoloration was observed.	Ozonated water reduced the yeasts, molds, aerobic mesophilic and psychrotrophic bacteria counts. On the other hand, chlorine was not effective to reduce aerobic mesophilic bacteria counts (for yeasts and molds and psychrotrophic bacteria, the best results were obtained when washing for 1 min).	Horvitz and Cantalejo, 2010 [81]
Broccoli	Aqueous ozone 2 μL L^−1^ for 90 s and 180 s	No significant differences were observed in gas composition and color parameters among different sanitizers (100 μL L^−1^ chlorinated water, electrolyzed water containing 100 μL L^−1^ free chlorine, 2 μL L^−1^ ozonated water) with contact times. No off-odor was detected during the storage.	A longer contact time was not effective in reducing microbial population, except with O_3_ washing. O_3_ with 90 s was not very effective in reducing microbial population compared with Cl or EW. However, samples washed with O_3_ for 180 s observed the lowest numbers of total aerobic and coliform plate counts.	Das and Kim, 2010 [82]
Green leaf lettuce	Aqueous ozone 2 ppm for 2 min	Ozone treatment showed no significant effect on ascorbic acid and d β-carotene contents and was found to be better than the chlorine and organic acid treatments in maintaining the sensory quality, compared with chlorinated water (100 ppm) and organic acid treatments.	No significant difference was observed between these three treatments in reducing the microbial load and controlling it during cold storage.	Ölmez and Akbas, 2009 [83]
Onion, escarole, carrot and spinach	Aqueous ozone 10, 20 and 80 mg min^−1^	Turbidity of wash water was reduced significantly by O_3_ and O_3_−UV treatments, while UV treatment did not affect the physicochemical quality of the water.	UV and O_3_−UV were effective disinfection treatments on vegetable wash water, with a maximum microbial reduction with O_3_−UV. However, maximum total microbial reductions were achieved by UV and O_3_ treatments, lower than by O_3_−UV treatment.	Selma et al., 2008 [84]
Iceberg lettuce	Aqueous ozone 4 mg L^−1^	No significant changes were observed in the texture and moisture content of lettuce samples dipped in chlorine, organic acids and ozonated water during storage. Color, β-carotene and ascorbic acid values of fresh-cut iceberg lettuce did not change significantly.	Organic acid dippings resulted in lower mesophilic and psychrotrophic counts than ozonated water and chlorine dippings.	Akbas and Olmez, 2007 [85]
Lettuce	Aqueous ozone 3.6 ppm	Through the addition of ozone to the wash water, the quality of lettuce during storage time was unaffected compared with water-washed lettuce.	Through the addition of ozone to the wash water, there was only a limited observed decrease in populations of microorganisms, compared with water-washed lettuce.	Hassenberg et al., 2007 [86]
Lettuce	Aqueous ozone 1 mg L^−1^	The use of ozone produced a significantly higher oxygen decline than the use of CLac (calcium lactate,15 g L^−1^). At the end of storage, CLac (alone or combined with ozone) samples had higher oxygen content (∼9%) than ozone samples (∼6%). Significant reductions in POD, PPO and enzymatic activity (polyphenol oxidase, peroxidase and pectin methylesterase activity) were observed in ozone samples. The reduction in pectin methylesterase activity has negatively affected textural properties.		Rico et al., 2007 [87]
Iceberg lettuce	Aqueous ozone 3, 5 and 10 ppm for 5 min	Ozonated water treatment increased the phenylalanine ammonia lyase (PAL) activity, compared with the water wash treatment, but the concentration of ozone did not affect PAL activity. Treatment with 3 or 5 ppm ozonated water resulted in more rapid changes in the a* value than after the water treatment. The ascorbic acid content of the lettuce was not affected by these treatments.	The native bacterial population on the lettuce declined in response to a rise in ozone concentration. However, there was no further bacterial reduction above 5 ppm ozone.	Koseki and Isobe, 2006 [88]
Celery	Aqueous ozone 0.03, 0.08, 0.18 ppm for 2, 6, 10 min, respectively	The polyphenoloxidase (PPO) activity and respiration rate was significant inhibited by treatment of ozonated water, and sensory quality of fresh-cut celery treated with ozonated water was better than that non-treated (at 0.18 ppm). There is no significant difference between ascorbic acid and total sugar of fresh-cut celery treated with ozonated water and non-treated.	Significant reduction of 1.7 log cycles of total microbial counts in fresh-cut celery treated with ozonated water at 0.18 ppm.	Zhang et al., 2005 [89]
Lettuce	Aqueous ozone 10, 20 and 10 mg L^−1^ min, activated by ultraviolet C (UV-C) light, in air or active modified atmosphere packaging (MAP)	Despite its strong oxidizing activity, ozonated water did not stimulate the respiratory activity of fresh-cut lettuce. Ozonated water maintained the initial visual appearance of fresh-cut lettuce and controlled browning during storage in air.	Initially, ozonated water and chlorine reduced the total mesophilic population by 1.6 and 2.1 log, respectively, when compared with water. Active MAP was effective in controlling total microbial growth, in relation to samples stored in air and caused a reduction in coliforms on sanitized samples compared with water-washed samples. The most efficient treatments were ozone 20 and ozone 10 activated by UV-C, which were as effective as chlorine.	Beltrán et al., 2005 [90]
Potato strips	Aqueous ozone 20 mg L^−1^ min	Under MAP, only sodium sulfite prevented browning, although it conferred off-odors, compared with other treatments (water, sodium hypochlorite, Tsunami, ozone and the combination of ozone–Tsunami). After 14 days of storage, there was no evidence of browning in fresh-cut potatoes dipped in ozonated water or ozone–Tsunami, and these treatments maintained initial texture and aroma, compared with other treatments.	The use of ozonated water alone was not effective in reducing total microbial populations. Ozone–Tsunami resulted in the most effective treatment to control microbial growth.	Beltrán et al., 2005 [91]

**Table 2 foods-10-00748-t002:** Effect on the quality parameters and microbiological control of fresh-cut fruits treated with aqueous ozone.

Fruits	Treatment	Qualitative Results	Microbiological Results	References
Apple	Aqueous ozone 1.4 mg L^−1^ for 5 min	Water-soluble pectin content increased more slowly, while protopectin content and cellulose content decreased at a lower rate in AO-treated fresh-cut apple compared with the control. AO treatment promoted increased pectin methylesterase activity and distinctly inhibited β-galactosidase and α-arabinofuranosidase activities during storage. Polygalacturonase activity was not affected by AO treatment.		Liu et al., 2020 [92]
Grapes	Aqueous ozone 2, 4, 6, 8 mg L^−1^	Ozonated water stimulated the respiration rate, especially after 5 days of storage, and increased superoxide dismutase and catalase activity compared to NaOCl (100 mg L^−1^) sanitized grapes. Total polyphenol content was 23–50% higher in Thompson Seedless (TS) and 18.5–28% higher in and Black (BS) samples sanitized with ozonated water. Twofold higher total antioxidant capacity (TAC) was registered in TS at all of the evaluated O_3_ doses while the doses of 6 and 8 mg L^−1^ increased TAC by 19–30% in BS. The use of ozonated water as a sanitizing method, especially at 6 and 8 mg L^−1^ doses, improved the functional quality.	The use of ozonated water as a sanitizing method, especially at 6 and 8 mg L^−1^ doses, maintained low microbial counts.	Silveira et al., 2018 [93]
Melon	Aqueous ozone 0.8 ppm	There are no adverse effects on SSC, color and firmness.	Reduction in total microbial counts on melon cubes (<2 log CFU g^−1^).	Botondi et al., 2016 [26]
Apple	Aqueous ozone 1.4 mg L^−1^ for 5 and 10 min	The ethylene production, polyphenol oxidase and peroxidase activities, and total phenol and malondialdehyde contents were reduced by aqueous ozone treatments. AO treatments delayed the quality deterioration and enhanced their antioxidant capacity.	AO treatment for 5 and 10 min achieved accepted microbial quality and, respectively, reduced total bacteria counts by 1.83 and 2.13 log_10_ CFUg^−1^ compared with the control samples.	Liu et al., 2016 [94]
Pineapple	Aqueous ozone 0.6, 0.9 and 1.5 ppm	The pH values of the ozone-treated samples were slightly but significantly higher than in control samples and also increased significantly over time in all samples. The quality parameters total soluble solids, ascorbic acid and total titratable acidity, color attributes and texture were not significantly different from those in the control samples. The microbial population was reduced as the ozone concentration increased.	The total plate count, total coliform and total yeast and molds were not significantly different from those in the control samples.	Nur Aida et al., 2011 [95]
Guava, pineapple and banana	Aqueous ozone 8 mL s^−1^ for 0, 10, 20 and 30 min	Total phenol and flavonoid contents of pineapple and banana increased significantly when exposed to ozone for up to 20 min, with a concomitant increase in ferric reducing antioxidant power (FRAP) and 2,2-diphenyl-1-picrylhydrazyl (DPPH) values. The opposite was observed for guava. Ozone treatment significantly decreased the ascorbic acid content of all three fruits.		Alothman et al., 2010 [96]

**Table 3 foods-10-00748-t003:** Effect on the quality parameters and microbiological control of fresh-cut vegetables treated with gaseous ozone.

Vegetables	Treatment	Qualitative Results	Microbiological Results	References
**Rocket**	Gaseous ozone 1, 2 and 5 ppm	Fresh-cut rocket was not adversely affected by the UV-C (5, 10 and 20 kJ m^−2^) and ozone (1, 2 and 5 ppm) treatment, maintaining the sensory quality during cold storage.	The 20 kJ UV-C m ^−2^ treatment was found to be better than the chlorine and gaseous ozone treatments, in terms of reducing the microbial load in fresh-cut rocket.	Gutiérrez et al., 2017 [97]
**Green Peppers**	Gaseous ozone 6.42 mg cm^−3^ for 15 min	The three treatments (ozone, modified atmosphere packaging and ozone + MAP) all reduced respiration rates and malondialdehyde (MDA) content compared to the control group. The enzyme activities in fresh-cut green peppers including peroxidase (POD), superoxidase dismutase (SOD) and L-phenylalanin ammonia-lyase (PAL) were induced by ozone and MAP treatments, while polyphenol oxidase (PPO) activities were inhibited.		Chen et al., 2016 [98]
**Lettuce, spinach and parsley**	Gaseous ozone 950 μL L^−1^, 20 min Aqueous ozone 12 mg L^−1^	AO does not affect chemical characteristics of the vegetables (chlorophyll a, chlorophyll b, ascorbic acid and total phenolic contents and antioxidant activity). GO caused significant losses in important bioactive compounds of parsley. Ascorbic acid and total phenolic contents and antioxidant activity in ozone-treated samples were 40.1, 14.4 and 41.0%, respectively, less than the control samples.	Aqueous ozone: chlorine and ozone washes resulted in average log units reductions of 2.9 and 2.0 for *E. coli* in the vegetables tested, respectively, while the efficiency of ozone (2.2 log) was very close to that of chlorine (2.3 log) on *L. innocua.* GO treatment resulted in 1.0–1.5 log reductions in the numbers of *E. coli* and *L. innocua* in parsley.	Karaca and Veliogu, 2014 [99]
**Bell peppers**	Aqueous ozone 1 μL L^−1^ Gaseous ozone 0.7 μL L^−1^	In all the experiments, O_2_ continuously decreased and CO_2_ concentration increased. The pH value increased and a significant softening was observed in all the fruits. By day 14, L values decreased in all the fruits, with the greatest changes found in the chlorinated samples (approximately 12 units). The aqueous solutions, ozonated (1 μL L^−1^) and chlorine (200 μL L^−1^) water showed greater changes in the quality attributes with increasing washing times. The GO treatment did not affect any of the physicochemical attributes studied.	The exposure for three min to gaseous ozone reduced the mesophiles, psychrotrophes and fungal populations of the fresh-cut peppers in 2.5, 3.3 and 1.8 log units, respectively, compared to ozone treatment in water.	Horvitz et al., 2014 [100]
**Tomato slices**	Gaseous ozone 0.4 mg L^−1^ for 1, 3 and 5 min	The poor appearance, aroma and overall quality obtained in all treatments.	It is recommended to wash tomato slices with 0.4 mg L^−1^ ozonated water for 3 min only. Extending treatment duration did not improve the microbiological quality, possibly due to the extra time permitting the ozone to react with other components of the fruit tissue, undermining the antimicrobial benefits.	Aguayo et al., 2006 [101]

**Table 4 foods-10-00748-t004:** Effect on the quality parameters and microbiological control of fresh-cut fruits treated with gaseous ozone.

Fruits	Treatment	Qualitative Results	Microbiological Results	References
Papaya	Gaseous ozone 9.2 μL L^−1^ for 10, 20 and 30 min	Following a 20 min ozone treatment, the total phenolic content of fresh-cut papaya increased by 10.3%, while the ascorbic acid content decreased by 2.3% compared to that of untreated control fruit.	Gaseous ozone reduced microbial counts being more effective on coliforms (0.39–1.12 log_10_ CFUg^1^) than on mesophilic (0.22–0.33 log_10_ CFUg^1^) bacteria.	Yeoh et al., 2014 [102]
Melon	Gaseous ozone 6.34 mg m^−3^	The integrity of the slices treated with gaseous ozone (GO) was preserved better than those of the others (hazelnut oil, NaClO), and no juice leakage was observed during storage. For sensorial attributes, control, NaClO and HO-treated melon slices preserved their quality for six days, whereas GO-treated samples were stored for nine days with good quality.	For microbial attributes, control, NaClO- and HO-treated melon slices were preserved their quality for six days, whereas GO-treated samples were stored for nine days with good quality.	Dilmaçünal et al., 2014 [103]
Cantaloupe	Gaseous ozone 5000, 20,000 and 10,000 ppm for 30 min	Gaseous ozone treatments maintained an acceptable visual quality, aroma and firmness.	Gaseous ozone 10,000 ppm for 30 min under vacuum reduced viable *Salmonella*. *Salmonella* viability loss was greater on dry exocarp surfaces than in the wetted surfaces during ozone treatment. Gaseous ozone treatment of 5000 and 20,000 ppm for 30 min reduced total coliforms, Pseudomonas fluorescens, yeast and lactic acid bacteria.	Selma et al., 2008 [104]

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
