# Peer review of "A Review into the Effectiveness of Ozone Technology for Improving the Safety and Preserving the Quality of Fresh-Cut Fruits and Vegetables"

_foods, 2021, doi:10.3390/foods10040748_

Round 1

Reviewer 1 Report

The authors present a well-written, comprehensive summary of the literature on the published research on ozone use to preserve and decontaminate fresh cut produce.

General

Lines 84-87 and throughout: Genus and species names (bacteria and plant) should be in italics and species names should be lowercase – including in references.

Lines 143-144 and throughout: Numbers in chemical formula should be subscript.

Line 267 and throughout: Squared and cubed numbers should be superscript.

Line 554, Tables, and throughout: What are the log units? cfu/g?

Line 79 and throughout: Expand acronyms on first use.

Specific

Lines 33-37: Although I agree generally, the current wording implies that “consumers acknowledge the need to include… phytochemical compounds, such as polyphenols, flavonoids, sterols, carotenoids, chlorophyll, anthocyanins in their daily diet”. I would suggest most consumers will not be familiar with these terms unlike “fiber”, “vitamins” and “minerals”.

Figure 2: Rising? Presumably ‘rinsing’.

Line 54: Are there more recent references than estimates from 2014?

Line 82: Water activity (aw) at first use

Line 88: Hepatitis A virus.

Lines 108-112: References?

Line 135: shelf life of sliced vegetables, presumably?

Line 145: different MAP component gas concentration?

Line 149-150: what kind of chemicals? O3 is a chemical.

Line 174: Quotation marks are unnecessary.

Line 217: Eliminates or inactivates?

Line 230: What constitutes ‘main studies’?

Tables 1-4: The referencing in the table is different to that in the rest of the manuscript making it hard to find the correct reference. Please make consistent.

Table 1: Expand for AO and GO on first use.

Table 1: Row 5 – “O157:H7” should not be italicised; Row 7 - Typhimurium is a Salmonella enterica serovar name and should not be italised. Row 8 – “Gram-negative”. Row 10 – ppm double up.

Table 2: Final row – only use of FRAP and DPPH. Expand.

Line 280: Ready-to-packaged?

Line 296: Expand on what L*, a*, b* and ‘hue angle’ mean when first used.

Line 456: Uniform use of vitamin C or ascorbic acid throughout. Or explain if they are being used interchangeably.

Line 550: Lactic not Lactate.

Line 571 and 578: ‘and’ instead of ‘e’.

Line 622: What constitutes a ‘type’ of fresh cut tissue? Can all RTE fruits/leafy vegetables/non-RTE foods be lumped together? How generalisable are data from each study?

Author Response

Please, see the attached file

Reviewer 2 Report

Development and detailed investigation of the efficiency of microbial decontamination methods for fresh fruit and vegetables can provide useful information not just for science but also for the industry practice. Review manuscript foods-1152431 focuses on the applicability for preserving of ozone technology for fresh-cut fruits and vegetables discussing of the change of quality parameters, as well. The manuscript is generlly well structured and well written. Introduction section summarized clearly the importance of ready-to-eat fruits and vegetables for consumers, their preservation, and sanitization technologies, and the principles and practice of ozone treatments (gaseous and ozonized water). Literature gap and research motivations are well defined. The manuscript contains interesting and valuable results based on relevant references. Effects of ozone treatments on technofunctional parameters, nutritional parameters and microbial parameters are discussed in details.

Suggestions:

I suggest the authors to discuss briefly the economy (cost) of ozone treatment in comparison with other microbial decontamination methods.

I suggest the authors to discuss the effects of ozone treatment on sensory properties in a separated section of the manuscript.

Author Response

Please, see the attached file
